



# Data Quality Enhancement for Atmospheric Chemistry Field Experiments via Sequential Monte Carlo Filters

Lenard L Röder[1], Patrick Dewald[1], Clara M Nussbaumer[1], Jan Schuladen[1], John N Crowley[1], Jos Lelieveld[1,2], and Horst Fischer[1]

[1]Max Planck Institute for Chemistry, Department of Atmospheric Chemistry, Mainz, Germany
[2]Climate and Atmosphere Research Center, The Cyprus Institute, Nicosia, Cyprus

**Correspondence:** Lenard Röder, lenard.roeder@mpic.de

**Abstract.** In this study we explore the applications and limitations of Sequential Monte Carlo filters (SMC) to atmospheric chemistry field experiments. The proposed algorithm is simple, fast, versatile and returns a complete probability distribution. It combines information from measurements with known system dynamics to decrease the uncertainty of measured variables. The method shows high potential to increase data coverage, precision and even possibilities to infer unmeasured variables. We extend the original SMC algorithm with an activity variable that gates the proposed reactions. This extension makes the algorithm more robust when dynamical processes not considered in the calculation dominate and the information provided via measurements is limited. The activity variable also provides a quantitative measure of the dominant processes. Free parameters of the algorithm and their effect on the SMC result are analyzed. The algorithm reacts very sensitively to the estimated speed of stochastic variation. We provide a scheme to choose this value appropriately. In a simulation study $O_3$, $NO$, $NO_2$ and $j_{NO_2}$ are tested for interpolation and de-noising using measurement data of a field campaign. Generally, the SMC method performs well under most conditions, with some dependence on the particular variable being analyzed.

## 1 Introduction

Insight into the complex chemical system of the atmosphere is often achieved by conducting coordinated field experiments where an ensemble of trace gases, meteorological variables, physical properties and aerosol compositions are measured that comprehensively characterize the sampled air masses (Hidalgo and Crutzen, 1977; Lelieveld et al., 2018; Wofsy et al., 2018). Field campaigns track variables along a spatio-temporal trajectory and are prone to local and temporal events. These are not resolvable by satellite measurements or chemical-transport models.

Quantitative analysis of data from field campaigns is often hindered by low data quality and insufficient data coverage of all parameters needed at each time step. The latter may result from poor instrumental time resolution, sporadic instrument failures, measurement duty cycles or instrument calibration. Assuming uncorrelated data loss of just $10\,\%$ per instrument, a field experiment with 10 different measurement instruments would lose $65\,\%$ of simultaneous data.

Reconstruction or enhancement of time steps with lost data or poor data quality is not easily achievable. Linear interpolation and moving average filters act as low pass filters that dampen high frequency variations of the measured variables. Thus, the




main advantage of the field measurement compared to remote sensing is suppressed. Calculation of missing data with photosta-
tionary state (PSS) calculations works for many species, but introduces a bias as all other processes are disregarded without an
estimate of reconstruction error (Ridley et al., 1992). Using the outputs of chemical-transport models as replacements lowers
the local and temporal resolution. The latter approach also contradicts with the goal of some field campaigns that try to evaluate
model predictions (Georgiou et al., 2018).

Sequential Monte Carlo (SMC) methods have become a useful tool in combining prior knowledge of a dynamical system
with noisy measurements. Originally applied to trajectory reconstruction (Kitagawa, 1996; Pitt and Shephard, 1999), this
method has become a major tool in meteorology for data assimilation to a theoretical model (Bauer et al., 2015; Van Leeuwen
et al., 2019). There SMC is applied to enhance the performance of a trusted model with additional information provided by
noisy measurements.

Ensemble Monte Carlo methods have been used in relation to atmospheric chemistry measurements where they enabled esti-
mation of dynamics (Krol et al., 1998), reactions (Berkemeier et al., 2017) or emission sources (Guo et al., 2009; Wawrzynczak
et al., 2013). Other novel applications cover enhancements of neural networks and machine learning methods (Doucet et al.,
2001; Ma et al., 2020).

The goal of this work is to explore the SMC method in the enhancement of data quality, data coverage and in the augmen-
tation of data to include unmeasured species in a system of measured atmospheric variables that are connected via known
chemical reactions. The focus is shifted from the enhancement of model outputs as in most recent studies (Van Leeuwen et al.,
2019) towards data quality enhancement as originally intended (Doucet et al., 2001). Enhanced data quality enables more com-
prehensive data analysis of field campaign measurement data. Several applications will be tested on measurements taken in
July 2021 at the Taunus Observatory, Kleiner Feldberg, Germany (Dewald et al., 2022). The study will focus on the chemical
system of ozone ($O_3$), nitric oxide (NO), nitrogen dioxide ($NO_2$) and the photolysis frequency of $NO_2$ ($j_{NO_2}$).

In the following section the basic theory of the SMC will be explained. In section 3 the underlying chemistry of the con-
sidered system and the measurement techniques used to derive the data-set will be described. In section 4 several experiments
using the measured data and SMC will be conducted and discussed.

## 2 Sequential Monte Carlo

The $N$-dimensional state vector of a system is defined as $\boldsymbol{x}_{t_n} =: \boldsymbol{x}_n \in \mathbb{R}^N$ at time steps $t_n$ ($n \in \{1..L\}$). This vector contains
all unknown or hidden true values in the system. This state vector evolves partly deterministically according to a transition
function $f_n$ and partly stochastically by addition of some noise $w_n$. The counterpart of $\boldsymbol{x}_n$ is the measurement vector $\boldsymbol{y}_n \in \mathbb{R}^M$
that contains all available measurements at time step $t_n$. The elements of the state vector and the measurement vector are
connected via an auxiliary function $h_n$ that depends on the state vector and measurement noise $v_n$. The functions $f$ and $h$ can
also depend on auxiliary parameters $\boldsymbol{u}$ that are considered to be exact.



## 2.1 Basic Procedure


The implementation of an SMC algorithm requires a known conditional probability distribution function (pdf) $p(\boldsymbol{y}_n|\boldsymbol{x}_n)$ that can be easily calculated and a procedure to sample from the prior pdf (Gordon et al., 1993)

$$p(\hat{\boldsymbol{x}}_n) = p(\boldsymbol{x}_n|\boldsymbol{x}_{n-1}). \tag{1}$$

The distribution $p(\boldsymbol{x}_n|\boldsymbol{x}_{n-1})$ contains prior information about the dynamics of the state vector and $p(\boldsymbol{y}_n|\boldsymbol{x}_n)$ describes the

probability of a measurement given a particular realization of the state vector $\boldsymbol{x}_n$. The latter probability distribution encodes the uncertainty of the measurement instruments.

Applying Bayesian theory the posterior pdf results from the calculation of the expression

$$p(\boldsymbol{x}_n) = p(\boldsymbol{x}_n|\boldsymbol{y}_n) = \frac{p(\boldsymbol{y}_n|\hat{\boldsymbol{x}}_n)p(\hat{\boldsymbol{x}}_n)}{p(\boldsymbol{y}_n|\boldsymbol{y}_{0:n-1})} \tag{2}$$

where $p(\boldsymbol{y}_n|\boldsymbol{y}_{0:n-1})$ depends on all previous information. Now the process can be considered *Markovian* as the dependence on

all previous information can be described by direct dependence on the most recent timestep only: $p(\boldsymbol{y}_n|\boldsymbol{y}_{0:n-1}) = p(\boldsymbol{y}_n|\boldsymbol{y}_{n-1})$. This distribution is most likely not a retractable expression (Doucet et al., 2000). An exception is the Kalman filter (Kalman, 1960) that requires the transition function $f$ and the measurement function $h$ to be linear and purely Gaussian distributed state and measurement noise. These conditions are not met in a system of chemical reactions as the reactions can be highly non-linear and abundances of molecules follow a probability distribution function (pdf) that is zero for negative abundances.

The main idea to overcome this numerical limitation in SMC is the approximation of the pdf of $\boldsymbol{x}_n$ with a finite amount of samples $\boldsymbol{x}_n^{(i)}$ (particles). The pdf of the state vector is approximated by the empirical distribution

$$p(\boldsymbol{x}_n) \approx \frac{1}{K}\sum_{i=1}^{K}\delta\left(\boldsymbol{x}_n - \boldsymbol{x}_n^{(i)}\right) \tag{3}$$

where $K$ represents the number of particles and $\delta$ denotes the Dirac measure. If the particles are sampled from the true pdf, the empirical pdf approximates the true pdf for $K \to \infty$. Given the initial particles were sampled according to $p(\boldsymbol{x}_0)$ this can

be ensured by sequential updating of the particles followed by a bootstrap filter. The particles are updated by application of the transition function to form the prior pdf:

$$\hat{\boldsymbol{x}}_n^{(i)} = f_n(\boldsymbol{x}_{n-1}^{(i)}, w_n^{(i)}) \sim p(\boldsymbol{x}_n|\boldsymbol{x}_{n-1}) \tag{4}$$

Then a weight is calculated for each particle that is related to the distance of the particle to the measurement:

$$q^{(i)} = \frac{p(\boldsymbol{y}_n|\hat{\boldsymbol{x}}_n^{(i)})}{\sum_{j=1}^{K}p(\boldsymbol{y}_n|\hat{\boldsymbol{x}}_n^{(j)})} \tag{5}$$

The posterior distribution is then approximated by bootstrap resampling (Gordon et al., 1993; Doucet et al., 2000) from the prior particles according to their individual weight $q^{(i)}$. This is implemented by calculation of the cumulative sum of all weights and





choosing the $k$-th particle where a uniform random variable $u^{(i)}$ is less or equal to the cumulative sum up to the $k$-th particle:

$$u^{(i)} \sim \mathcal{U}(0,1) \tag{6}$$

$$\boldsymbol{x}_n^{(i)} = \hat{\boldsymbol{x}}_n^{(k)} \quad \text{where } u^{(i)} \leq \sum_{j=1}^{k} q^{(j)} \tag{7}$$

$$p(\boldsymbol{x}_n) \approx \frac{1}{K} \sum_{i=1}^{K} \delta(\boldsymbol{x}_n - \boldsymbol{x}_n^{(i)}) \tag{8}$$


## 2.2 Auxiliary Particle Filter

This approximation assumes that $K$ is large enough. The main challenge of the method is the possibility of the particles to collapse into a single mode (Snyder et al., 2008). It is possible that a single particle carries a weight very close to 1 while all other particles carry weights close to 0. In these cases the posterior approaches a $\delta$-distribution without any statistics. In the

literature there are many approaches to counter this problem, maintaining similar weights for all particles (Van Leeuwen et al., 2019). This is especially important for high dimensional problems such as data assimilation, as the number of particles has to grow exponentially with the size of the measurement vector $M$ (Snyder et al., 2008). A common approach (Doucet et al., 2000; Van Leeuwen et al., 2019) suggests sampling from a proposal distribution $q(\boldsymbol{x}_n|\boldsymbol{y}_{0:n})$ instead of the prior, that nudges the particles into the direction of the posterior before applying the bootstrap filter. Pitt and Shephard (1999) described a method

they called *Auxiliary Particle Filter* where they define the proposal pdf as

$$q(\boldsymbol{x}_n^{(i)}|\boldsymbol{y}_{0:n}) = p(\boldsymbol{\mu}_n^{(i)}|\boldsymbol{y}_n) \tag{9}$$

where $\mu$ is a likely draw from the prior pdf. This can be achieved by a simplified version of the transition function $f_n$ without a stochastic part. For each particle an intermediate weight $\lambda^{(i)}$ is calculated and afterwards $R > K$ samples are drawn from the particles where

$$p(j = i) \propto \lambda^{(i)} \quad \text{with} \quad j \in \{1..R\}. \tag{10}$$

The prior is then constructed from this mixture prior analogous to Eq. (4). For the posterior evaluation the weights (5) have to be rescaled by the first stage weights (10) to compensate the introduced bias before applying the bootstrap filter (7). This method can still lead to weight collapse but increases the statistics and efficiency of the SMC, as the particles of the posterior empirical pdf are less likely to be degenerate by construction (Van Leeuwen et al., 2019). Recently Fearnhead and Künsch

(2018) and Van Leeuwen et al. (2019) reviewed several novel approaches to counter this problem for high dimensional systems such as weather forecasting; Pulido and van Leeuwen (2019) proposed a particle flow formalism that completely counters weight collapse (Hu and van Leeuwen, 2021).

These methods will not be considered in more detail, however, since they are typically dealing with $N \sim 10^9$ and $M \sim 10^7$, while a measurement field campaign lies in the range $N, M < 100$. Due to the high dimension of the former systems, it is very

likely that several measurements differ from the prediction by many standard deviations. As mentioned above the goal in those





cases is to optimize a trusted model with the information provided via noisy measurements. In our case the centerpiece of the system is the observation. The results from our SMC algorithm should never disagree with the measurements but should rather assist the observations; finding a more precise estimate of a variable, similar to a weighted average of several measurements.

Further weight maintaining adaptions to the SMC can be considered for future applications. For now weight collapse will
be tracked throughout the experiments as a metric. In this study the entropy

$$H(\hat{\boldsymbol{x}}_n) = -\sum_{i=1}^{K} \lambda_n^{(i)} \log\left(\lambda_n^{(i)}\right) \tag{11}$$

$$H(\boldsymbol{x}_n) = -\sum_{i=1}^{K} q_n^{(i)} \log\left(q_n^{(i)}\right) \tag{12}$$

will be considered. $H$ is close to its maximum value $\log(K)$ or $\log(R)$, respectively, when all particles share similar weights. The maximum value is reached if and only if the measurement does not contribute any additional information. The effective
dimension $R^*$ of a posterior can be approximated by $\exp(H)$ where a total collapse to a single particle corresponds to $H \to 0$ and $R^* \to 1$. Low entropy is not necessarily a tracer for poor performance of the SMC method but might indicate vast deviations of the actual chemical system from the considered model. An example might be sudden emission of relevant trace gases, changes in wind direction or other local effects.

## 3   Chemical Reactions and Measurements

This study focuses on the interplay between tropospheric $O_3$, NO and $NO_2$. According to Leighton (1961) and Nicolet (1965) the concentrations of these trace gases reach a steady state a few minutes during daytime. The relevant reactions are

$$O_3 + NO \quad \rightarrow \quad O_2 + NO_2 \tag{R1}$$

$$NO_2 + h\nu \quad \rightarrow \quad NO + O \tag{R2}$$

$$O + O_2 + M \quad \rightarrow \quad O_3 + M \tag{R3}$$

where reaction R3 can be considered fast compared to R2. The reaction coefficient $k_{O_3,NO} =: k_1$ is taken from Atkinson et al. (2004). The photolysis frequency $j_{NO_2}$ varies between circa zero at night and several $10^{-3}\text{s}^{-1}$. The photostationary state is reached when

$$k_1 \, [O_3] \, [NO] = j_{NO_2} [NO_2]. \tag{13}$$

Under atmospheric conditions, this photostationary state is additionally affected by peroxy radicals predomininanty orig-
inating from the oxidation of volatile organic compounds (VOCs) by e.g. OH or $O_3$. Both hydroperoxy ($HO_2$) and organic peroxy radicals ($RO_2$) convert NO to $NO_2$ (R4 and R5). In addition, further chemical reactions, direct emission, deposition and transport processes influence this steady state (Crutzen, 1979; Parrish et al., 1986; Ridley et al., 1992).

$$NO + HO_2 \quad \rightarrow \quad NO_2 + OH \tag{R4}$$

$$NO + RO_2 \quad \rightarrow \quad NO_2 + RO \tag{R5}$$





The coordinated measurements (TO21 campaign) took place in July and August 2021 on the mountain Kleiner Feldberg (826m, 50° 13' 18" N, 8° 26' 45" O), Germany, located in a rural, forested region under anthropogenic influence from several large cities within a radius of circa 35km. This site has been used before for field campaigns and is described in more detail in Crowley et al. (2010) and Sobanski et al. (2016). NO and $NO_2$ were measured via a photolysis-chemiluminescence detector described in Tadic et al. (2020) and Nussbaumer et al. (2021). $j_{NO_2}$ was calculated from actinic flux measurements by a spectral

radiance detector (MetCon GmbH) (Bohn and Lohse, 2017). Ozone was measured by two commercial UV absorption monitors (2B Technologies). Several of other trace gases and chemical variables were measured during the campaign that will not be considered in this study. An in depth discussion of all measurements in this campaign can be found in Dewald et al. (2022). Meteorological data were provided by a weather station of the German Weather Service (DWD) on the summit.

## 3.1   SMC Setup

The setup used in this study is based on the following definition: The state vector $\boldsymbol{x}$ and the measurement vector $\boldsymbol{y}$ are both four-dimensional and encode the mixing ratio of $O_3$, NO and $NO_2$ in units of parts ber billion volume (ppbv) and the photolysis frequency $j_{NO_2}$ in $s^{-1}$. Therefore, the auxiliary function $h$ simplifies to the identity. The transition function $f$ is composed of an initial randomization of each dimension that follows a lognormal distribution and numerical integration of the differential equation resulting from the chemical reactions. The parameters of the distribution are chosen so that mean and standard

deviation are equal to the current value and a given standard deviation $\boldsymbol{\sigma}_0$, respectively. The choice of a lognormal distribution for chemical systems has been discussed e.g. in Limpert et al. (2001) and solves the problem of otherwise possible negative values for the abundances. With the scheme proposed here, the lognormal distribution approximates a Gaussian distribution as the standard deviation becomes smaller than the mean. The reactions R1 and R2 result in the differential equation

$$\frac{d}{dt}\boldsymbol{x} = \begin{pmatrix} -k_1\ [O_3]\ [NO]\ \frac{p}{k_B T} + j_{NO_2}\ [NO_2] \\ -k_1\ [O_3]\ [NO]\ \frac{p}{k_B T} + j_{NO_2}\ [NO_2] \\ -j_{NO_2}\ [NO_2] + k_1\ [O_3]\ [NO]\ \frac{p}{k_B T} \\ 0 \end{pmatrix} \tag{14}$$

where $k_B$ is Boltzmann's constant that converts the reaction coefficient to units of $ppbv^{-1}s^{-1}$ at given pressure $p$ and temperature $T$. These values will be input into the calculation as auxiliary variables for each time step. The parameters $p$, $T$ and $k_1$ will be set as fixed values and their uncertainties will not be considered. However, any deviations from the true values can be compensated via the initial randomization. The function used to calculate the weights as in Eq. (5) is defined as the product of Gaussian kernels

$$p(\boldsymbol{y}_n | \hat{\boldsymbol{x}}_n) \propto \prod_{m=1}^{M} \exp\left( -\frac{(\hat{\boldsymbol{x}}_{n,m} - \boldsymbol{y}_{n,m})^2}{2\sigma_{n,m}^2} \right) \tag{15}$$

where $\sigma_{n,m}$ is constructed by

$$\sigma_{n,m}^2 = DL_m^2 + (P_m \hat{\boldsymbol{x}}_{n,m})^2 \tag{16}$$



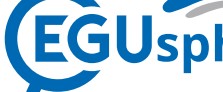

using the characteristic detection limit DL and precision P for each instrument. In cases of missing measurements the factor is set to 1. In the actual algorithm Eq. (15) is replaced by

$$\log\left(p(\boldsymbol{y}_n|\hat{\boldsymbol{x}}_n)\right) = \sum_{m=1}^{M} -\frac{(\hat{\boldsymbol{x}}_{n,m} - \boldsymbol{y}_{n,m})^2}{2\sigma_{n,m}^2} \tag{17}$$

to ensure numerical stability.

The model output is constructed from full Bayesian inference to convert the approximate probability distribution to an estimate for the state and an estimate of the error:

$$\overline{\boldsymbol{x}}_n = \mathbb{E}_{\boldsymbol{x}_n \sim p(\boldsymbol{x}_n)}[\boldsymbol{x}_n] \approx \frac{1}{K}\sum_{i=1}^{K} \boldsymbol{x}_n^{(i)} \tag{18}$$

$$\Delta\boldsymbol{x}_n = \sqrt{\mathbb{V}_{\boldsymbol{x}_n \sim p(\boldsymbol{x}_n)}[\boldsymbol{x}_n]} \approx \left(\frac{1}{K-1}\sum_{i=1}^{K}(\boldsymbol{x}_n^{(i)} - \overline{\boldsymbol{x}}_n)^2\right)^{\frac{1}{2}} \tag{19}$$

## 3.2 Model Extension

In each iteration the individual particles will evolve towards the photostationary state where Eq. (14) equals zero. This state corresponds to the photostationary state for a given $NO_x = NO + NO_2$, $O_x = O_3 + NO_2$ and $j$. Particles which are close to the measurement then have high probability to be sampled. If no measurement is available all particles are equally likely to be

sampled. Therefore the posterior will shift towards the photostationary state in the unsupervised state. This behavior can cause high biases in combination with low uncertainty on the prediction if no measurements are available and, at the same time, the chemistry is dominated by processes not regarded in the algorithm. During nighttime the photolysis frequency of $NO_2$ is zero so other sources of NO e.g. emissions from soil or plants play a dominant role (Wildt et al., 1997). An example of this effect can be found in the supplement.

Thus, we extend the state vector described in the previous section with an additional variable $\eta \in \{0,1\}$ so that $\boldsymbol{x}^* = (\boldsymbol{x}, \eta)$. This variable will be called *activity* and gates the differential equation according to

$$\frac{d}{dt}x^* = \eta \frac{d}{dt}\boldsymbol{x}. \tag{20}$$

Now each particle can be either *active* or *passive* if $\eta = 1$ or $0$, respectively. If the chemical reactions incorporated into Eq. (14) dominate the chemistry, it is more likely that *active* particles survive and vice versa. A small probability $p_\eta$ to switch activity

is included into the randomization phase to prevent mode collapse. This way the algorithm can turn chemical processes on and off, whichever is more likely according to the measurements. Additionally, the mean value of $\eta$ can give insights into the relative importance of the chemical processes incorporated.

Figure 1 shows the diel profile of $\eta$ for the complete dataset in comparison with the diel average of $j_{NO_2}$, along with a Box-Whisker plot. For low photolysis frequencies the activity lies close to zero and sharply increases with higher actinic flux.

At $j \approx 0.003$ s$^{-1}$ the median activity rises above $50\%$ and later saturates at around $90\%$. The $10\%$ and $25\%$ quantiles suggest a very skewed distribution at noon. This indicates deviations from the PSS calculation due to other processes e.g. reaction R4.



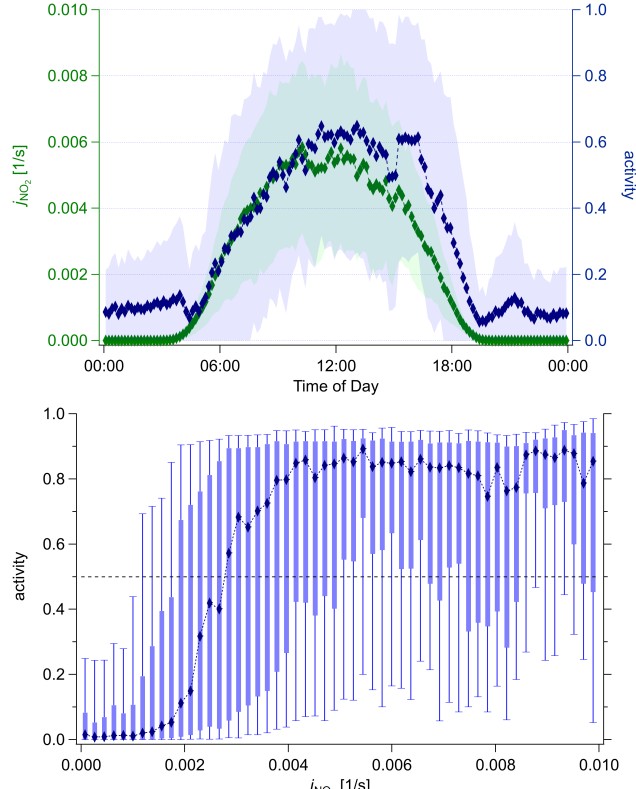

**Figure 1.** Top: diel profile of photolysis frequency $j_{NO_2}$ (green) and activity $\eta$ (blue) averaged for each minute interval. Time is in UTC. Bottom: Box-Whisker plot of activity as a funtion of photolysis frequency. The markers and dotted line marks the median, boxes range from the 25 % quantile to 75 % quantile, the whiskers mark the 10 % and 90 % quantiles, respectively. The black dashed line marks the transition from passive to active regime $\eta > 0.5$. A clear non-linear correlation is visible. Although the correlation is non-linear the Pearson correlation coefficient equals 0.67.

### 3.3 Comparison with Constrained Box-Model Calculations

Similar calculations have been conducted using observationally constrained box-models (Hens et al., 2014; Crowley et al., 2018; Dewald et al., 2022) in which a selection of measured parameters (e.g. trace gas mixing ratios and photolysis rates) are used as time-dependent inputs for a detailed chemical reaction scheme. Physical effects e.g. deposition and uptake can be adjusted to best replicate measured outputs (Dewald et al., 2022). Sensitivity studies can be conducted by variation of reaction rates and other parameters (Crowley et al., 2018). Typically these model calculations have runtimes in the order of seconds for a full dataset, dependent on the number of reactions.

From a qualitative perspective these calculations have some similarities with the SMC. However, in our case the choice of appropriate constraints is based on Bayesian theory and the quantitative measurement uncertainty. Sensitivity studies are automatically obtained due to the description of the state as a probability distribution. Unconsidered effects can be compensated





via stochastic variability. Also, measurement errors are not directly propagated to the output since the measurement vector is separated from the state vector. In the limit of low constraint uncertainties and full chemical description of the system, the outputs of SMC and box-model calculations converge. In other cases the latter may be used to prepare a full SMC run, benefiting from its low runtime, and enables detailed chemical investigation of the system (Crowley et al., 2018).

## 4 Experiments

In order to study the effect of the SMC method on time series of chemical systems, several experiments were conducted on the measurement. The capability of the method to interpolate missing data points was tested by artificially discarding data and comparing the reconstruction of the SMC algorithm with the original measurement. The result is evaluated using the mean square error (MSE), and the squared error divided by the standard deviation ($\chi^2$):

$$\mathrm{MSE} = \frac{1}{L} \sum_n (\overline{\boldsymbol{x}}_n - \boldsymbol{y}_n)^2 \qquad (21)$$

$$\chi^2 = \frac{1}{L} \sum_n \left( \frac{\overline{\boldsymbol{x}}_n - \boldsymbol{y}_n}{\Delta \boldsymbol{x}_n} \right)^2 \qquad (22)$$

This will be described in more detail in sections 4.1 and 4.2. Another possible application is enhancement of the precision of a measurement within a system. For this test white noise is added to the observed data. The reconstruction is also analyzed in terms of MSE and $\chi^2$. The model is performing well if the MSE is close to the uncertainty of the measurement and $\chi^2$ is close to 1. Finally we discuss the possibility to augment the dataset to include unmeasured variables.

We give a depiction of the algorithm used in sections 4.1 and 4.2 in Algorithm 1.

### 4.1 Interpolation

The SMC method is tested as an alternative to interpolation of missing data by randomly discarding sections of data with interval size $T$. This process is repeated for each dimension of the state vector i.e. each molecule and the photolysis frequency. Then the missing data is reconstructed.

The algorithm described at the beginning of this section is applied to the whole dataset. Missing data in each dimension is automatically interpolated since the algorithm returns a value for $\boldsymbol{x}$ at all time steps. The uncertainty is given by the standard deviation of the ensemble of particles (19). If data is missing in some dimension, the likelihood of the particles is less sparse. This leads to survival of more particles and a higher spread of the posterior. Hence, the standard deviation and the entropy increase. Once there is measurement data available again, only data points close to the measurements will be sampled. Entropy and standard deviation decrease again. If the mean of the distribution strongly deviates from the measurement at this point, only few particles survive and both entropy and standard deviation become very small. The resulting standard deviation underestimates the uncertainty of the model at this point. The requirements for the approximation of the posterior with the finite sample of particles does not hold any more. Therefore, data points where the entropy is small will be discarded. This threshold




---

**Algorithm 1** Auxiliary Particle Filter in a $O_3$, NO, $NO_2$, $j_{NO_2}$, $\eta$ system.

---

**Require:** $\boldsymbol{x} = (O_3, NO, NO_2, j_{NO_2}, \eta)$

  **for all** timesteps $t_n$ **do**

    **Auxiliary phase**

    $\boldsymbol{\mu}^{(i)} \leftarrow \boldsymbol{x}_{n-1}^{(i)} + \dot{\boldsymbol{x}}_{n-1}^{(i)} \Delta t$ using (14),(20)

    Calculate $\lambda^{(i)} \leftarrow p(\boldsymbol{y}_n|\boldsymbol{\mu}^{(i)})$ using (17)

    Sample $R$ random particles $\boldsymbol{x}^{(j)}$ from $\boldsymbol{x}_{n-1}^{(i)}$ weighted by $\lambda^{(i)}$

    **Randomization phase**

    **for all** particles $\boldsymbol{x}^{(j)}$ **do**

      **for all** $\xi \in \{O_3, NO, NO_2, j_{NO_2}\}$ **do**

        Resample $\xi$ from lognormal distribution with mean $\xi$ and standard deviation $\boldsymbol{\sigma}_{0,m}$

      **end for**

      Switch $\eta$ to $1-\eta$ with probability $p_\eta = 0.025$

    **end for**

    $\boldsymbol{x}^{(j)} \leftarrow \boldsymbol{x}^{(j)} + \dot{\boldsymbol{x}}^{(j)}\Delta t$ using (14), (20)

    Calculate $q^{(j)} \leftarrow p(\boldsymbol{y}_n|\boldsymbol{x}^{(j)})$ using (17)

    Rescale by auxiliary weights $q^{(j)} \leftarrow \frac{q^{(j)}}{\lambda^{(i)}}$

    Sample $K$ random particles $\boldsymbol{x}^{(i)}$ from $\boldsymbol{x}^{(j)}$ weighted by $q^{(j)}$

    $\boldsymbol{x}_n^{(i)} \leftarrow \boldsymbol{x}^{(i)}$

  **end for**

---

is set to

$$H(\boldsymbol{x}_n) < \log(K) \Leftrightarrow R^* < K \tag{23}$$

as there are effectively less particles left to sample from than samples to be drawn.

    Figure 2 shows an example plot of NO with random artificial data gaps of 30 min that have been interpolated by the SMC.

Depending on activity, the SMC ensemble mean either tends towards the PSS equilibrium or stays approximately constant, while the ensemble spreads and increases the standard deviation. This spreading happens fast in the beginning of a data gap and slows down afterwards. This might follow the $\sqrt{N}$ behavior of a sum of normally distributed variables. In this example plot the spreading speed matches the behavior of the system so that the measurement at the end of a data gap lies within the $\pm 1\sigma$ interval.

This procedure was repeated for each species and a wide range of data gaps between 1 minute and 1 day. The data gaps were shuffled 8 times for each setup to achieve better statistics. Figure 3 shows the resulting MSE. For low gap sizes the MSE stays constant. This constant corresponds to the base deviation of the SMC estimate from the unaltered measurement. This value is expected to be larger than zero, since the model combines the prior knowledge with the measurement and therefore introduces a small bias. We will call this bias *intrinsic model bias*.





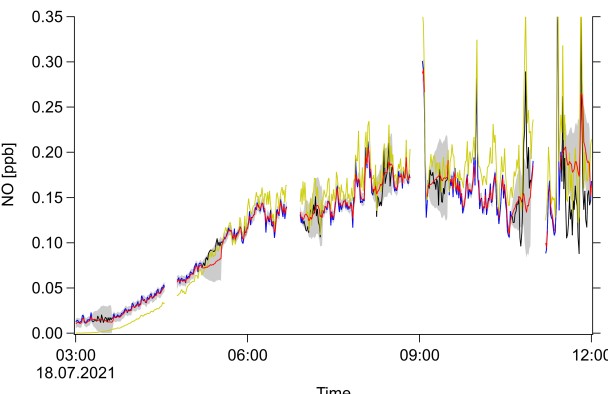

**Figure 2.** Example result of SMC used for interpolation. NO mixing ratio as a function of time (UTC) for an arbitrary day of the field campaign. Original measurement (black), measurement with artificial gaps (blue), PSS calculation (yellow), SMC ensemble mean (red) and $\pm 1\sigma$-interval (shaded grey region).

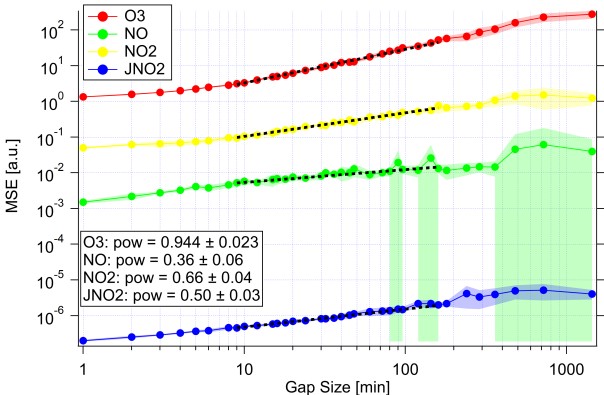

**Figure 3.** MSE of the SMC estimation as a function of artificial data gap size to study the interpolation capabilities. Mean MSE as lines and markers and standard deviation as shaded region for the ensemble of repetitions. The plot shows the results of all variables ozone (red), NO (green), $NO_2$ (yellow) and $j_{NO_2}$ (blue). Note the unit of MSE is arbitrary to fit all variables in one plot. The unit is $\mathrm{ppbv}^2$ for the MSE of trace gases and $\mathrm{s}^{-2}$ for the MSE of the photolysis frequency. The dashed black lines show power-law fits ($y = Ax^{\mathrm{pow}}$) fitted to the intermediate regions. The fit estimate for pow is given in the annotation.

With increasing gap size the MSE starts to increase. In Figure 3 it is clearly visible that the slope varies strongly with different variables. This slope corresponds to the power-law coefficient of MSE with larger gap size. In the uninformative case of linear interpolation and brownian noise, this slope is equal to 1. A lower power-law coefficient indicates an effective contribution of information through the remaining measurements considered. This coefficient is close to 1 for ozone, therefore this method is not capable of estimating the course of ozone in cases of instrument failure considering this particular system. This is not a

strong limitation since ozone can be measured precisely enough with commercial instruments. However, one should keep this





limit in mind in other systems where a species cannot be effectively described by the chemistry of the remaining variables in the system.

At very high gap sizes the MSE jumps to higher values and the standard deviation also increases. This indicates a higher sensitivity to the particular data gap position. In the limit the SMC estimate approaches the PSS calculation since no additional
information can be provided via measurements. For the variables that can be estimated by PSS reasonably well the MSE does not increase anymore at the largest gap sizes.

The second performance measure $\chi^2$ also increases with larger gap size, but the slope is more sensitive to the actual dynamics of the dataset. If the PSS gives a reasonable estimate of a value but at the same time conflicts with another important process, the SMC estimate follows the PSS and predicts a very low standard deviation. The actual measurement can be multiple standard
deviations away from the SMC estimate, thus a high value of $\chi^2$ is reached. For NO and $j_{\mathrm{NO_2}}$ this is most likely the case due to an additional NO source from soil and an additional NO sink via reaction R4. Here $\chi^2$ reaches high values before the two hour mark. A plot can be found in the supplement.

## 4.2 Precision Enhancement

In this section the SMC is applied to artificially noised measurements to test the capability to reconstruct the original signal. The
SMC combines the prior knowledge given by the system dynamics and the precisely measured variables with the remaining information provided by the noisy measurement. If the prior overlaps with the likelihood, the result will be a more precise estimate of the noised variable. If the prior is far away from the measurement due to another process dominating the system, e. g. during the night, the posterior will be close to the likelihood. An example plot in shown in Figure 4. Here normally distributed noise is applied to the $NO_2$ measurement. The expected different effects at daytime and nighttime are clearly visible. During
the night, the SMC result follows the structure of the measurement while PSS outputs unrealistic values. The noise is reduced only by regularization of the variability through $\sigma_0$. This effect will be discussed in more detail later. At daytime the SMC estimate lies between measurement and PSS calculation with a strong tendency towards the photostationary state.

Algorithm 1 is again applied to the noised datasets to obtain values for MSE, $\chi^2$ and also the baseline MSE. The latter value in fact equals the square of the noise added. Figure 5 shows the results of this experiment. In all cases the MSE is constant
for low noise. It is dominated by the intrinsic model bias. With increasing noise the MSE starts to rise once the baseline MSE reaches the intrinsic model bias. At this point, though, the MSE increases less steeply than the baseline MSE. Therefore, the additional information provided by the system dynamics successfully decreased the noise. At the same time the value for $\chi^2$ starts to increase. The SMC becomes overly confident as the prediction according to the dynamics can no longer be falsified by measurement accuracy. MSE and $\chi^2$ start to saturate when the limit of extrapolation is reached.
The value of $\chi^2$ of the photolysis frequency decreases at the beginning, until the added noise gets close to the detection limit. Up to this point the increased uncertainty during the night influences the uncertainty of the SMC estimate which decreases $\chi^2$. For $O_3$ the system starts to diverge at high uncertainty. Again this indicates that ozone cannot be completely reconstructed from the PSS calculation using only the considered molecules. Thus, the application of the method to de-noising is limited when other processes dominate.



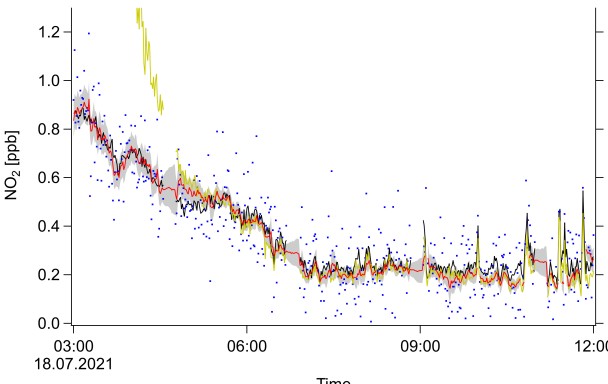

**Figure 4.** Example result of SMC used for denoising. $NO_2$ mixing ratio as a function of time (UTC) for an arbitrary day of the field campaign. Original measurement (black), measurement with artificial noise (blue), PSS calculation (yellow), SMC ensemble mean (red) and $\pm 1\sigma$-interval (shaded grey region).

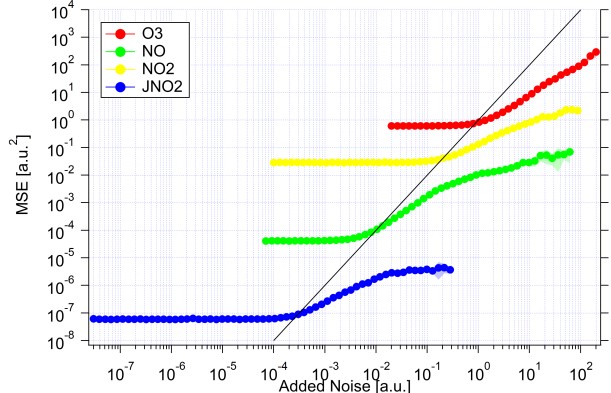

**Figure 5.** MSE of the SMC estimation as a function of artificial noise to study the precision enhancement abilities. Mean MSE as lines and markers and standard deviation as shaded region for the ensemble of repetitions. The plot shows the results of all variables ozone (red), NO (green), $NO_2$ (yellow) and $j_{NO_2}$ (blue). Note the units of MSE and artificial noise are arbitrary to display all variables in one plot. The unit of the noise is ppbv for the trace gases and $s^{-1}$ for the photolysis frequency. The unit of MSE is the square, respectively. The solid black line indicates the baseline MSE ($MSE = noise^2$)

## 4.3 Extrapolation

The state vector can also be appended with an unmeasured variable. If this variable is strongly coupled to measured variables through the system dynamics the SMC can give reasonable estimates. This problem can also be interpreted as the limit of infinitely large data gaps or measurements with infinite uncertainty.



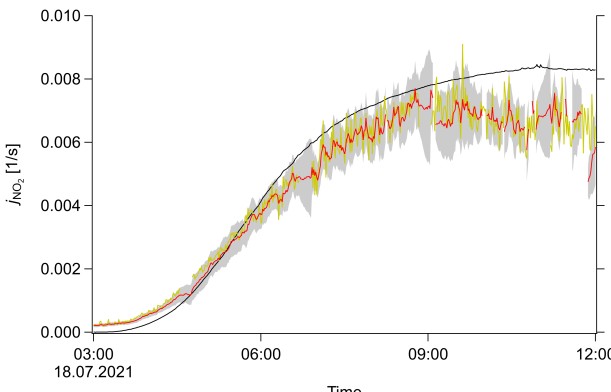

**Figure 6.** Example result of SMC used for inference. Photolysis frequency $j$ as a function of time (UTC) for an arbitrary day of the field campaign. Original measurement (black), PSS calculation (yellow), SMC ensemble mean (red) and $\pm 1\sigma$-interval (shaded grey region).

Figure 6 shows an example plot of $j_{\mathrm{NO_2}}$. The SMC result matches with the photostationary state calculation, as expected. Additionally, the SMC is aware of the actual speed of the chemical reactions and is regularized via $\sigma_0$ with regards to the speed of unconsidered effects. In the example plot the estimation shows only moderate agreement with the actual measurement. This discrepancy can be explained by other effects interfering with the system e.g. NO emission from soil during the nighttime or other sinks of NO such as R4 and R5.

One has to be careful, however, if multiple unknown variables are coupled. In the case that a small variation in one can be compensated by variation of another variable, the system is singular and will most likely diverge to unrealistic values within a few iterations.

### 4.4 Free System Parameters

The performance of the SMC can change under variation of important free parameters. The most basic parameter is the measurement error $\sigma_{n,m}$ that relates to the detection limit DL and precision P via Eq. (16). This parameter governs how far particles are allowed to spread from the measurement. An overestimation of this error will bias the algorithm output towards the prior estimate, an underestimation will bias the output towards the measurement. However, $\sigma_{n,m}$ can easily be chosen appropriately if the values of DL and P match the actual performance of the instrument during the measurement.

The switching probability $p_\eta$ has to be tuned for a reasonable performance. If $p_\eta$ is too high, activity is not dominated by inheritance but random switching. The algorithm will output values similar to the system where activity is set to 1, but with slightly slower dynamics enabled. This also leads to unstable behavior. If $p_\eta$ is too small, it is hard to switch from one state to another. Therefore, the algorithm will become unstable if the environmental conditions switch too fast. Examples are given in the supplement, for the interpolation of the photolysis frequency.

The standard deviation of the prior $\boldsymbol{\sigma_0}$ has to be chosen carefully. It encodes the expected speed of variation of the system due to stochastic and unconsidered effects. This regularizes the resulting time series to low frequencies. If an appropriate value




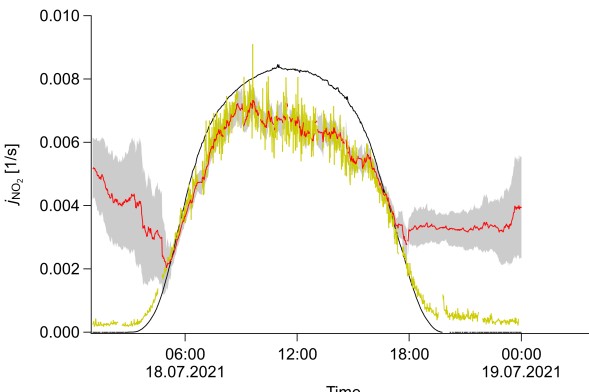

**Figure 7.** Variation of the free parameter $\sigma_0$ and its exemplary effect on the inference of $j_{NO_2}$. $\sigma_{0,\mathrm{rel}}$ is decreased by a factor of 3. Photolysis frequency $j$ as a function of time (UTC) for an arbitrary day of the field campaign. Original measurement (black), PSS calculation (yellow), SMC ensemble mean (red) and $\pm 1\sigma$-interval (shaded grey region).

is chosen for this standard deviation, the algorithm already shows nice denoising characteristics for each individual variable in $\boldsymbol{x}$ even if no system dynamic is considered at all. This effect has been reported e.g. by Riris et al. (1994) and Leleux et al. (2002) for applications of a simple Kalman filter to mixing ratios of trace gases. Therefore, one might encounter seemingly good performance of this algorithm when in fact the result is just dominated by the expected speed of variations.

   If the value is chosen too small, the system cannot reproduce rapid changes that do not originate from the chosen dynamic.
Figure 7 shows an example plot for the photolysis frequency. The system cannot catch up with the speed of sunrise and sunset and decouples from the measurement. If the value is chosen too high, the standard deviation is increased. This can lead to a flattened out probability distribution that effectively reduces the statistics of appropriate particles and therefore can also lead to unstable behavior.

   Here we propose the analysis of the entropy as a measure. If $\sigma_0$ is too small, the distribution is very condensed and all
particles get similar weights. The entropy approaches a constant value. If the value is too high, the distribution spreads out and particles at the edge of the distribution get much lower weights than particles in the center. The entropy decreases with increased $\sigma_0$. A proper choice of this parameter lies in the transition region from constant entropy to decreasing entropy. An example plot can be found in the appendix, Figure A8.

   Throughout this study $\boldsymbol{\sigma_0}$ is calculated for each step from a constant and a linear contribution similar to Eq. (16), where
$\sigma_{0,\mathrm{const}}$ and $\sigma_{0,\mathrm{rel}}$ are obtained from a linear fit of the measured difference between consecutive samples vs the measurement itself. This choice falls into the transition region mentioned before. Additional elaborations with regards to this parameter as well as the used $\sigma_0$ values can be found in the appendix.





## 5 Conclusions

In this study, we demonstrate that the SMC is a very versatile method that can effectively enhance data quality of atmospheric
field measurements. We have shown satisfactory results when applied to data coverage increase, precision enhancement and
inference of unmeasured variables. The algorithm is composed of simple steps and only introduces simplified chemical dy-
namics into a system of measurements. This way the data quality can be enhanced without precise knowledge of complex
reactions and processes such as emission, uptake, deposition or mixing with other air masses. The algorithm automatically
detects deviations from the proposed simple dynamics by switching from the active state to the passive state. This ensures
stability and gives quantitative insights about the underlying dominant processes. Furthermore, the entropy value encodes the
information gained through the measurement and therefore the missing information in the prior estimate.

Along with several benefits over other approaches we also explored the limitations of this method. Without the model
extension by the activity variable $\eta$ the algorithm can produce unrealistic estimates when the system dynamics deviate from
the proposed reactions. Variables which follow the proposed dynamics quite well and only differ slightly will lead to an
underestimation of the standard deviation. Variables which do not follow the proposed dynamics at all do not benefit from the
system dynamics but will be regularized with regards to the speed of possible variations. In this case the algorithm is very
sensitive to the proposed values of $\sigma_0$. This free parameter has to be chosen very carefully. However, a proper value can be
chosen by the analysis of observed variations and entropy.

The proposed method should not be seen as a replacement for PSS calculations, box-model calculations, model estimates
or actual measurements, but is an extension to the arsenal of numerical analysis for atmospheric chemistry measurements. It
provides many desirable properties as it is very simple, returns salvageable higher moments of the estimated distribution while
requiring a low runtime. A single run with the described setup and the whole 32 days dataset took 18 minutes of runtime on an
8-core desktop PC.

In general we emphasize the versatility and high potential of this algorithm. Under the right circumstances, SMC can be
utilized to enhance data quality and data coverage to allow for a more comprehensive data analysis of field campaign measure-
ment data. However, we suggest to conduct similar experiments when applied to a new system of variables. In particular, if the
method is applied to a system of precise measurements along with a single imprecise, irregular or nonexistent measurement,
the latter variable should be analyzed with regards to interpolation capability, precision enhancement ability and sensitivity to
hyper parameters before conclusions can be drawn from the SMC result. These tests could be conducted on modeled data or
on a different dataset where the same variables were measured.

*Code and data availability.* Data of the TO2021 campaign are available upon request to all scientists agreeing to the data protocol at Crowley
et al.. Python code is published on github: https://github.com/lenroed/smc-boxmodel.





*Author contributions.* LR initiated the study, carried out the calculations and analysis and wrote the manuscript. CN, PD and JS provided measurement data. JC and PD contributed to the chemical interpretation of the dataset. JL and HF supervised and consulted the study and
defined the goals of this paper.

*Competing interests.* We declare no competing interests.

*Acknowledgements.* This work was supported by the Max Planck Graduate Center with the Johannes Gutenberg-Universität Mainz (MPGC). We thank Andreas Kürten and Joachim Curtius (Institute for Atmospheric and Environmental Sciences, Goethe University, Frankfurt am Main), for the logistical support and access to the facilities at the Taunus Observatory. We thank the DWD, for the provision of meteorological
data.





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
