# Peer review of "Data Quality Enhancement for Atmospheric Chemistry Field Experiments via Sequential Monte Carlo Filters"

_EGUsphere, 2022_

## Referee Comment (RC2)

The manuscript introduced sequential Monte Carlo filters (SMC) to atmospheric chemistry field experiments and demonstrated its capability of enhancing the measured data qualities as well as estimating unmeasured variables. The paper is well-written, and the context is within the scope of AMT. I recommend minor revision before publication if the comments below are addressed.

1. My major concern is the robustness of SMC algorithm when applied to a more complicated system. The study only uses one experiment with 5 dimensions (O3, NO, NO2, JNO2, and gate variable). How does the performance of this algorithm change as the dimension of the system increases?

2. Introducing activity variable as a regularization term prevent the system from mode collapse. However, it is unclear how $p(\eta)$ is chosen even though it is discussed in section 4.4.

3. Line 285-290: Please specify that the results of $\chi^2$ is shown in Figure S3.

4. Line 326: Figure A8 should be Figure S8

---

## Author Response (AR1)

Dear Anonymous Referee #2,

we thank you very much for your time and effort spent to provide your comments.

1. Why is the nighttime PSS value in Fig4 so abnormal? Is it the problem of box model simulation?"

In order to address this question, some details of the calculation need to be considered. In PSS the time derivative of the system is assumed to be zero, so the production and loss terms are equal. For NO$_2$ this results in $j_{NO_2}[NO_2] = k_1[NO][O_3]\frac{p}{k_B T} \Rightarrow [NO]_2 \sim \frac{[NO][O_3]}{j_{NO_2}}$. So [NO2] is inversely proportional to the photolysis frequency, which tends to zero during nighttime. This leads to an unstable behavior of PSS in absence of sunlight.

2. Why is the ensemble mean value of SMC so different from the PPS calculation at 12 o'clock in Fig6?

We investigated this and found multiple occurrences in the extrapolation example where the ensemble becomes unstable and entropy is reduced. This is also visible in the plot in Figure 6 where the red curve is disconnected due to discarded points as mentioned in 4.1. We added this discussion to section 4.1 (L 235ff):

This threshold is set to
$$H(\overrightarrow{x_n}) < \log(K) \Leftrightarrow R^* < K$$

as there are effectively less particles left to sample from than samples to be drawn. After a low entropy incident, the ensemble may require a few iterations to converge again. Considering this effect and discarding additional points may increase the data accuracy while lowering data coverage. Throughout this analysis no additional points were discarded. In applications the amount of low entropy events may be reduced using an increased ensemble size which requires a longer runtime.

Dear Anonymous Referee #1,

we thank you very much for your time and effort spent to provide your comments.

1. My major concern is the robustness of SMC algorithm when applied to a more complicated system. The study only uses one experiment with 5 dimensions (O3, NO, NO2, JNO2, and gate variable). How does the performance of this algorithm change as the dimension of the system increases?

We originally thought about including increased dimension sizes. Unfortunately, a detailed analysis as shown in the article for the 5-dimensional case was not possible for an increased dimension size for any of the datasets available. All considered data sets featured limitations in data coverage and quality that limit the significance of this analysis when the number of dimensions is increased.
The system NO, NO2, O3, j_NO2 is particularly simple and is not very sensitive to other trace gases in such a tropospheric environment. However, increasing the dimensionality to include variables like HOx, VOCs or peroxides quickly introduces many reactions. Applying SMC to such systems may yield great chemical insights, similar to constrained box-model studies. However, this would shift the scope of the study from a measurement technique perspective (AMT) to application and chemical analysis (ACP), as no "ground truth" benchmark can be provided.
Our conclusion is that the SMC needs many studies of different chemical systems in the future to fully rate its potential. This study provides a first step by testing limitations on a relatively limited dataset.
We added the following sentences to the conclusion regarding the system dimension (L 354ff):

An open question is the stability of the algorithm when applied to a more complicated system with higher dimension. Repeating the technical procedure of this study using a higher-dimensional system is restricted by data coverage and data quality in existing datasets. We suggest many applications of this method for different chemical systems are necessary in the future to fully rate the potential of the SMC in the analysis of atmospheric chemistry field experimental data.

2. Introducing activity variable as a regularization term prevent the system from mode collapse. However, it is unclear how $p(\eta)$ is chosen even though it is discussed in section 4.4.

The term $p_\eta$ corresponds to a value between 0 and 1 used for a binomial experiment. The value corresponds to the probability that the state is switched from active to passive or vice versa. As mentioned in Algorithm 1 or Table S1, the value is chosen to be 2.5% during the study. This value seemed a reasonable choice after applying the considerations discussed in section 4.4.

3. Line 285-290: Please specify that the results of $\chi^2$ is shown in Figure S3.

Thank you for this hint, the text might be unclear without this information. We added a sentence (L285).

A similar plot showing the resulting values of $\chi^2$ is shown in the supplement, Figure S3. The value of $\chi^2$ of the photolysis frequency decreases at the beginning, [...]

4. Line 326: Figure A8 should be Figure S8

Thank you for your note, we further changed all mentions of "appendix" to "supplement" accordingly. (Lines 328 and 332)